# Ghrelin as a Biomarker of Stress: A Systematic Review and Meta-Analysis

**DOI:** 10.3390/nu13030784

**Published:** 2021-02-27

**Authors:** Jean-Baptiste Bouillon-Minois, Marion Trousselard, David Thivel, Brett Ashley Gordon, Jeannot Schmidt, Farès Moustafa, Charlotte Oris, Frédéric Dutheil

**Affiliations:** 1CNRS, LaPSCo, Physiological and Psychosocial Stress, Université Clermont Auvergne, F-63000 Clermont-Ferrand, France; jschmidt@chu-clermontferrand.fr (J.S.); fdutheil@chu-clermontferrand.fr (F.D.); 2Emergency Medicine, CHU Clermont-Ferrand, F-63000 Clermont-Ferrand, France; 3Neurophysiology of Stress, Neuroscience and Operational Constraint Department, French Armed Forces Biomedical Research Institute (IRBA), F-91223 Brétigny-sur-Orge, France; marion.trousselard@gmail.com; 4Laboratory of the Metabolic Adaptations to Exercise under Physiological and Pathological Conditions [AME2P], Université Clermont Auvergne, F-63000 Clermont Ferrand, France; david.thivel@uca.fr; 5Holsworth Research Initiative, La Trobe University, Bendigo, 3083 VIC, Australia; b.gordon@latrobe.edu.au; 6CNRS, INSERM, GReD, Université Clermont Auvergne, F-63000 Clermont-Ferrand, France; fmoustafa@chu-clermontferrand.Fr (F.M.); coris@chu-clermontferrand.fr (C.O.); 7Biochemistry and Molecular Genetic Department, CHU Clermont-Ferrand, F-63000 Clermont-Ferrand, France; 8Occupational and Environmental Medicine, CHU Clermont-Ferrand, WittyFit, F-63000 Clermont-Ferrand, France

**Keywords:** appetite, anxiety, metabolism, public health, mental health

## Abstract

Introduction: Ghrelin is an orexigenic hormone which favors food-seeking behavior and has been postulated to be a biomarker of stress. We conducted a systematic review and meta-analysis on the evolution of ghrelin levels following acute stress. Methods: The PubMed, Cochrane Library, Embase, and ScienceDirect databases were searched for studies reporting ghrelin levels before and after acute stress in humans. Results: We included ten studies for a total of 348 patients. Acute stress (intervention) was always in a laboratory. Acute stress was psychological (Trier Social Stress Test), physical, or mixed (cold pressure test). The overall meta-analysis demonstrated an increase in ghrelin after the stress intervention (ES = 0.21, 95CI 0.09 to 0.34) compared with baseline levels. Stratification by time demonstrated an acute increase in ghrelin levels in the five minutes immediately following the initiation of stress (0.29, 0.10 to 0.48) but without any difference after. Obese individuals had a more significant (ES = 0.51, 95CI 0.18 to 0.84) and prolonged increase in ghrelin levels for up to 45 min compared with non-obese individuals who had a significant increase only five minutes after stress. Moreover, the ghrelin levels increased in response to stress with BMI (coefficient 0.028, 0.01 to 0.49; *p* = 0.013) and decreased with the time after the stress intervention (coefficient -0.007, −0.014 to −0.001; *p* = 0.025). Conclusion: Ghrelin is a biomarker of stress, with a short-term increase following acute stress. Obese individuals have both a higher and prolonged response, emphasizing the link between obesity and stress.

## 1. Introduction

Psychosocial stress has been known as a major public health problem for more than 30 years [1]. Identifying acute stressful events with objective measures is growing in interest in physiology and preventive medicine [2]. Biomarkers of stress may help to assess specific working conditions and develop preventive strategies for stress management that can be evaluated. Despite the most common biomarkers of stress derived from the hypothalamic–pituitary–adrenal axis being associated with the physiological stress response [3], other biomarkers have been proposed. Ghrelin is a 28-amino-acid peptide discovered in 1996, most commonly known as one of the main appetite regulation hormones, with an orexigenic role [4]. Increased ghrelin levels influence meal initiation and food-seeking behavior [5]. However, besides its multiple functions and despite conflicting results, ghrelin has been suggested to be a biomarker of stress [4], due to increased levels following acute stress. Under normal physiological conditions, ghrelin exhibits high and immediate variations in response to a meal [6], with a short circulating half-life of less than half an hour [7,8]. Ghrelin has an acute, immediate, and non-prolonged response to a stimulus, suggesting that the increase in ghrelin levels in response to acute stress may have a short duration. However, the results are equivocal concerning the ghrelin response to acute stress. To our knowledge, no meta-analysis to date has examined the effects of acute stress on ghrelin levels. Moreover, stress and obesity are intrinsically linked [9]. Stress has been suggested to lead to obesity via inappropriate eating behaviors [10]. Therefore, we hypothesized that obese individuals will exhibit a larger response in ghrelin levels following acute stress. Understanding the possible impact of ghrelin in the biological relationship between obesity and stress may help to determine an efficient preventive strategy. 

Thus, we aimed to conduct a systematic review and meta-analysis to demonstrate that ghrelin could be a biomarker of acute stress, with a short-term increase following acute stress, and that the ghrelin response may be greater in obese individuals compared with normal-weight individuals.

## 2. Materials and Methods

### 2.1. Literature Search

We searched PubMed, Cochrane Library, Embase, and ScienceDirect with the following keywords, “ghrelin” and “stress”, on 15 September 2020, for all articles describing our primary outcome variable—i.e., the measurement of ghrelin levels before and after acute stress—with or without a control group. We chose to use broad keywords to retrieve all possible articles. The search was not limited to specific years, languages, or regions. Additionally, reference lists of all publications that met the inclusion criteria were manually searched to identify any further studies that were not found with the electronic search, as well as reference lists from reviews. We excluded studies on animals and patients with psychiatric disorders. Two authors (J.-B.B.-M. and M.T.) conducted all literature searches, collated the abstracts, and separately reviewed the abstracts, and based on the selection criteria, decided the suitability of the articles for inclusion. A third author (F.D.) was asked to review the articles where consensus on suitability was debated. All the authors then examined the eligible articles. The search strategy is presented in Figure 1.

### 2.2. Quality of Assessment

We used the Newcastle Ottawa Scale (NOS) [11] to evaluate the quality of the included articles. The NOS was developed to assess the quality of studies based on three types of biases: bias of selection, bias of comparability, and bias of exposure. Those three items were evaluated through eight items or sub-items, with one point given when the study fulfilled the criteria. The maximal score was eight, which was then converted into a percentage.

### 2.3. Statistical Considerations

The statistical analysis was conducted using Stata software (version 16, StataCorp, College Station, TX, USA). The ghrelin levels were summarized for each study sample and reported as the mean and the standard-deviation (SD) before and after the acute stress. Random effects meta-analyses (DerSimonian and Laird approach) were conducted when data could be pooled. *p* values less than 0.05 were considered statistically significant. We described our results by calculating the effect size (ES, standardized mean differences as SMD) of ghrelin levels from before and after acute stress. A positive ES denotes an increase in ghrelin levels. A scale for ES has been suggested with 0.8 reflecting a large effect, 0.5 a moderate effect, and 0.2 a small effect. In particular, we conducted three main meta-analyses stratified by time after the withdrawal of the stress: a global meta-analysis with all participants, a meta-analysis with only normal-weight individuals, and a meta-analysis with only obese individuals. All the articles included reported several measurement times of ghrelin levels following acute stress. All the meta-analyses were stratified by time after stress. The number of stratifications was determined by the number of studies within each stratification. For example, the main meta-analysis was stratified on four time measures: <5 min, 5–15 min, 15–45 min, and >45 min. Heterogeneity in the study results was evaluated by examining forest plots, and confidence intervals (CI), and by using formal tests for homogeneity based on the I² statistic, which is the most common metric for measuring the magnitude of between-study heterogeneity and is easily interpretable. I² values range between 0% and 100% and are typically considered low at <25%, modest at 25–50%, and high at >50%. For rigor, funnel plots of these meta-analyses were used to search for potential publication biases. In order to verify the strength of the results, further meta-analyses were then conducted, excluding studies that were not evenly distributed around the base of the funnel. We also tried to compute a meta-analysis stratified on the type of sampling (blood, saliva, etc.) or the type of stress.

Meta-regression analyses were conducted to explore the influence of the study or the participants’ characteristics on the standardized mean differences. The following characteristics were considered when available: age, sex of the participants (male versus female), body mass index (BMI) and other sociodemographic variables, characteristics of the acute stress (duration, type of stress), and the time of sampling. Results were expressed as regression coefficients and 95%CI.

## 3. Results

The initial search retrieved 6631 putative articles (60 in the Cochrane Library, 651 in PubMed, 82 in Embase, and 6723 in ScienceDirect). The removal of duplicates and the use of the selection criteria reduced the number of articles reporting the evaluation of ghrelin level in blood or saliva to ten articles for the systematic review [12,13,14,15,16,17,18,19,20,21] and nine for the meta-analysis—with one study not reporting the number of participants [15] (Figure 1). All all articles were written in English.

### 3.1. Quality of Articles and Study Designs

Using the NOS criteria, all of the ten included studies had a score of >6 and considered high-quality (Figure 2). All the studies were cohort studies without blind assessment and mentioned ethical approval.

### 3.2. Inclusion and Exclusion Criteria

All of the participants were adults. One study included participants who were physically active or involved in competitive sports [12], two recruited only overweight women [20,21], two recruited obese and normal-weight women [14,18], and five recruited healthy individuals [13,15,16,17,19], men [13,16] and women [15,16,17] (Table 1). Participants were recruited via advertisements sent to University campuses [12,13,15], from an online poll [17], from the local newspaper [14,20,21], and from a clinic [18]; two studies did not specify recruitment procedures [16,19]. Most studies [13,14,15,16,17,18,19,20,21] shared the same exclusion criteria based on clinical eating disorders (in the past or present), acute medical illnesses, a regular medications that could influence hormone levels, and current or prior substance-related or other psychiatric disorders. Two studies with overweight individuals excluded those with recent dieting or an unstable body weight—i.e., a variation of 5% over the past three months [20,21]. One study did not describe the exclusion criteria [12].

### 3.3. Population

Sample sizes ranged from 9 [19] to 85 [14] (Table 1). We included a total of 305 patients. Ages ranged from 19.3 ± 2.0 years [17] to 50.2 ± 12.1 years [18]. One study did not specify the mean age, but only that age was between 18 and 30 [14]. Sex was reported in all studies: seven studies recruited only women [13,14,15,16,17,20,21], one recruited both men and women [18], and two only men [12,19]. In total, 22% of the included participants were men. Body Mass Index (BMI) ranged from 21.2 ± 2.0 [18] to 35.5 ± 4.6 kg/m^2^ [20]. No studies reported smoking, alcohol consumption, or leisure physical activity.

### 3.4. Outcome and Aim of the Studies

The main objective of all of the included studies was to assess the effect of acute stress instigated in a laboratory on ghrelin levels (and other hormones).

### 3.5. Type of Stress

Seven studies used the Trier Social Stress Test [13,14,15,16,17,18,19]. This procedure combines social stress (public speech] with mental stress (arithmetic under time pressure) and is a validated tool to provoke psychobiological stress responses. Two studies used the Cold Pressor Test [20,21], i.e., the immersion of the non-dominant hand up to the wrist in a rectangular-shaped container of 0 °C ice-water for 2 min. One study used a physical stressor (maximum oxygen uptake treadmill running test) [12] (Table 1).

### 3.6. Method of Sampling for Markers Analysis

Nine studies measured ghrelin levels in blood [12,13,14,15,17,18,19,20,21] and one in saliva [16] (Table 1). All the studies included in the meta-analysis gave details on the method of sampling and analysis. The blood samples were collected in EDTA treated tubes and immediately centrifuged to yield plasma for hormone determinations. The plasma was frozen between −20 °C [12] and −80 °C [19] until laboratory analysis. Concentrations were determined by radio-immuno-assay (RIA) [12,17,18,19,20,21] or enzyme-linked-immunosorbent-assay (ELISA) [13,14]. All studies also reported an inter-assay and intra-assay coefficient, ranging from 3.2% [20,21] to 12% [18], and from 1.91% [13] to 8% [17], respectively. 

### 3.7. Meta-Analyses of Ghrelin Variation Depending on Time after Acute Stress

The overall meta-analysis of nine studies (41 groups) demonstrated an increase in ghrelin levels after the stress intervention (ES = 0.21, 95CI 0.09 to 0.34) compared with the baseline levels. Stratification by time demonstrated an increase in ghrelin levels in the 5 min following the stress intervention (0.29, 0.10 to 0.48) but there was no statistical difference between 5 and 15 min (0.22, −0.04 to 0.48), between 15 and 45 min (0.40, −0.08 to 0.88), and after 45 min (0.04, −0.18 to 0.26) (Figure 3).

### 3.8. Meta-Analyses in Normal-Weight Population Were Based on Eight Studies (28 Groups)

Stratification by time demonstrated an increase in ghrelin levels in the 5 min following the stress intervention (0.23, 0.01 to 0.45) but without a difference between 5 and 45 min (0.10, −0.09 to 0.28) and after 45 min (0.02, −0.20 to 0.24), nor in overall results (0.11, −0.01 to 0.23) (Figure 4).

### 3.9. Meta-Analyses in Overweight Individuals Based on Four Studies (15 Groups) 

Stratification by time demonstrated an increase in ghrelin levels in the 5 min following the stress intervention (0.52, 0.03 to 1.02), between 5 and 45 min (0.71, 0.15 to 1.27), and in the overall results (0.51, 0.18 to 0.84). The only non-significant difference was after 45 min (0.19, −0.44 to 0.82) (Figure 5).

### 3.10. Meta-Regressions and Sensitivity Analysis 

Levels of ghrelin after stress increased with BMI (ES = 0.026, 95CI 0.004 to 0.48). Ghrelin levels decreased with time after the stress intervention (−0.007, −0.014 to −0.001). Sociodemographic variables (age, sex) did not influence the level of ghrelin, nor did the duration of stress or time of sampling (Figure 6). A sensitivity analysis demonstrated similar results after the exclusion of studies that were not evenly distributed around the funnel plot (Figure 7). Insufficient data precluded further analyses by type of stress (mental, physical, or both) or by type of sampling (blood, saliva, etc.).

## 4. Discussion

The main findings were that ghrelin is a biomarker of stress, with a short-term increase following an acute stress intervention. Moreover, overweight and obese individuals had an extended response compared to normal-weight individuals, demonstrating the link between obesity and stress.

### 4.1. Ghrelin as a Biomarker of Stress

The active form of ghrelin (acyl-ghrelin) is a 28-amino-acid peptide discovered in 1996 [22], with multiple functions [23] but is mainly an appetite-regulating hormone produced by the stomach [24] whish acts on the central nervous system [25]. Very interestingly, we demonstrated that acyl-ghrelin is also a biomarker of stress that increases following exposure to a stress environment. Acyl-ghrelin is known for two main effects: (1) balancing energy by promoting food intake [26,27] and secreting of insulin [28] and ACTH, and (2) release of growth hormone (GH) [23,29]. The increase in ghrelin in response to stress can be an adaptive mechanism that may help people with stress [30]. Moreover, high ghrelin levels seem to influence the learning process and memory [31,32]. From a biomolecular point of view, ghrelin is the result of two cleavages from the 117-amino-acids pre-pro-ghrelin [33] followed by an octanoylation by the Ghrelin O-Acyltransferase (GOAT) to produce the active form of ghrelin (acyl-ghrelin). Despite demonstrating an increase in active ghrelin levels following acute stress, the mechanisms of the increase are not yet known, i.e., an increase of the production of the pre-pro-ghrelin, an increase of the cleavages, and/or an increase of the GOAT octanylation. Lastly, contrary to common thoughts, ghrelin and GOAT are not only expressed in the stomach [25,34]. They are also expressed in different tissues such as the pancreas, small-intestine, anterior pituitary gland, placenta, lymphocyte, kidney, gonads, lung, brain, and hypothalamus [24,25]. Hypothalamus expression is mediated by the circadian clock and environmental factors such as stress.

### 4.2. Acute Response to Acute Stress Intervention

We demonstrated that ghrelin levels had a short-term increase in response to acute stress, with high levels immediately after the stressful event and then a gradual decrease. This could be explained in part by its short half-life of between 10 and 31 min [7,8]. Therefore, ghrelin can be considered a biomarker of acute stress, such as catecholamines, heart rate variability, or cytokines [35,36]. Moreover, ghrelin is also implicated in the regulation of stress through its action on the hypothalamic-pituitary-adrenal axis [30]. Physiologically, the main role of ghrelin is for meal initiation and food-seeking behavior [6]. Daily variations of plasma ghrelin levels show transient increase in pre-prandial periods followed by a decrease in post-prandial periods [37,38]. The decrease has been proposed to be triggered by absorption of nutrients within the intestine [39]. However, the decrease in plasma ghrelin levels following acute stress cannot be linked with meal stimulus, so other mechanisms of regulation should apply. Ghrelin also has a short-term role in the regulation of mental health; it is implicated in short-term adaptations against depression, and in the control of anxiety after acute stress [30].

### 4.3. Ghrelin Levels and Characteristics of Stress

In our meta-analysis, ghrelin levels globally increased with whichever the type of stress was involved (physical, mental, or both). Even if there was no statistical difference between the types of stress, the literature is scarce regarding the exploration of ghrelin levels during physical stress [12] and both physical and mental stress [20,21]. Most of the studies included in our meta-analysis used the (Trier Social Stress Test) TSST, one of the most common tools to induce acute mental stress. Even though a dose-response relationship has been demonstrated between the duration or the type of stress and some biomarkers of stress, we failed to demonstrate a dose-response relationship between the duration of stress and ghrelin levels. Moreover, none of the nine included studies reported such relations. A limited number of studies with physical stress precluded further analyses. However, studies of other biomarkers suggest a larger increase when both physical and mental stress is induced [20]. In our meta-analyses, as well as in literature, sociodemographic variables (age, sex) did not influence ghrelin levels. However, larger increases in cytokines have been reported in older individuals under stressful conditions [40]. Lastly, even if insufficient data precluded further analyses, blood sampling is commonly reported as being a better way to assess stress compared with saliva or urine sampling [41].

### 4.4. A prolonged Response in Overweight and Obese Individuals

Even though a positive effect size was identified for the overall population, there is a difference between normal-weight and overweight people. Indeed, the normal-weight population’s effect size was small and only significant during the first five minutes. This could impact our conclusions on the overall meta-analysis and possibly reduce the generalization of our results. In contrast, we showed a higher and longer increase in ghrelin levels in the overweight patients than in the normal-weight patients, in accordance with the literature [42]. Moreover, we demonstrated that levels of ghrelin after stress increased with BMI. Stress and obesity are two public health issues [43,44] that are intrinsically linked [9]. It has been proposed that stress can lead to obesity via inappropriate eating behaviors [10] through a non-adaptive response to ghrelin levels [9]. In addition, obesity is a major stress factor [45]. Stressed people are those who have the greatest difficulties losing weight [46]. The relationship between obesity and stress is biological via the main stress hormones regulating appetite (leptin, ghrelin) [47,48]. This relationship between obesity and stress facilitated an international recommendation that suggests implementing stress management programs in obesity for a sustainable weight loss [49]. The direction of the relationship between the heightened and prolonged ghrelin response to stress in overweight and obese individuals requires further consideration. It is not yet possible to determine if a higher ghrelin response to stress contributes to the development of obesity, or if obese individuals have developed a heightened ghrelin response as a result of weight gain. The results from our studies therefore have implications to use ghrelin as a screening tool to identify people at risk of obesity. Although associations between ghrelin levels, obesity and stress exist, causal pathways have not been established [50,51]. The biological pathways of ghrelin response to acute stress are yet to be fully understood [52], and therefore the biological explanations of a greater ghrelin response following acute stress in obese individuals are also lacking.

### 4.5. Limitations

Our study has some limitations. Noon of the included studies had a randomized controlled design. Moreover, blinding interventions were deemed infeasible. Though there were similarities between the inclusion criteria, they were not identical. Limiting our meta-analysis to studies sharing precisely the same inclusion and exclusion criteria was not possible due to the limited data. All of the studies were monocentric, limiting the generalizability of our results. However, our selection of articles was rigorous, and few studies were outliers considering meta-funnels. The low I^2^ to the attested homogeneity between the studies. Moreover, the homogeneity of the data precluded further sensitivity analysis of the different methods for measuring ghrelin levels (ELISA and RIA), stressors (TSST, sport, and CPT), and sampling (blood and saliva). The high I^2^ (69.1%) for meta-analysis in obese individuals might limit the generalization. However, we demonstrated a positive effect of BMI in our meta-regression.

## 5. Conclusions

Ghrelin is a biomarker of stress, with a short-term increase following acute stress. Obese individuals have both a higher and prolonged response, emphasizing the link between obesity and stress. Appetite regulation testing warrants further investigation for use as part of weight management strategies, especially in overweight/obese individuals.

## Figures and Tables

**Figure 1 nutrients-13-00784-f001:**
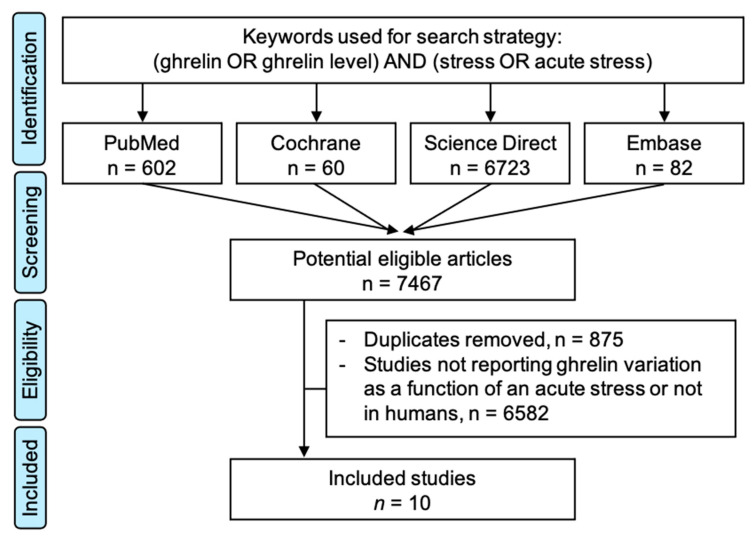
Search strategy.

**Figure 2 nutrients-13-00784-f002:**
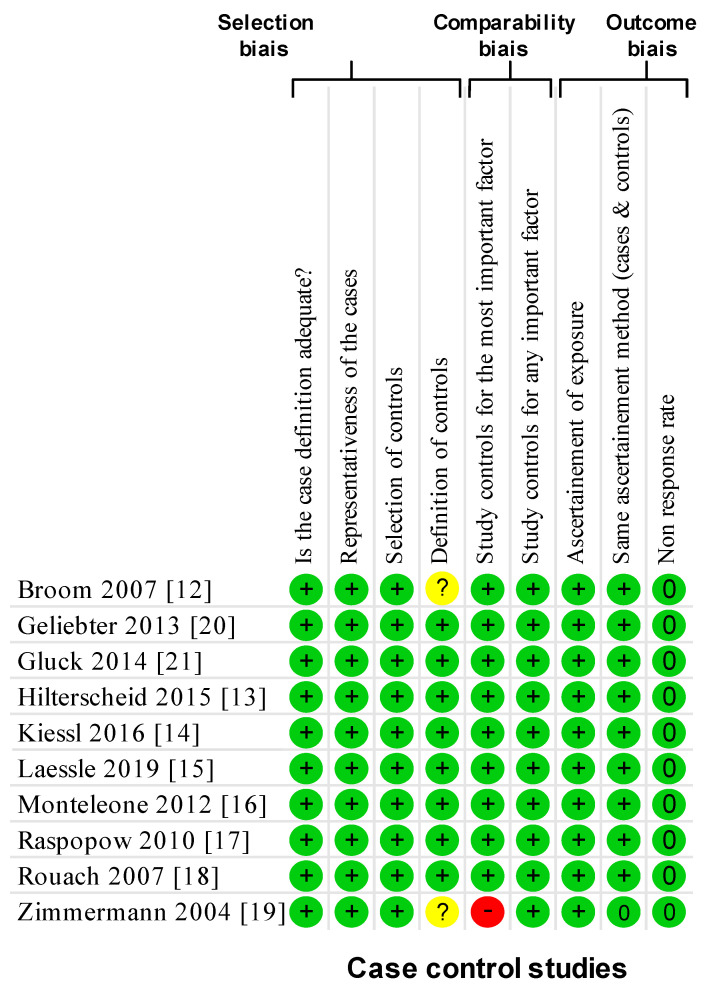
Methodological quality of included articles using the Newcastle–Ottawa Quality Assessment Scale. Yes: +; no: -; cannot say: ?; not applicable: NA.

**Figure 3 nutrients-13-00784-f003:**
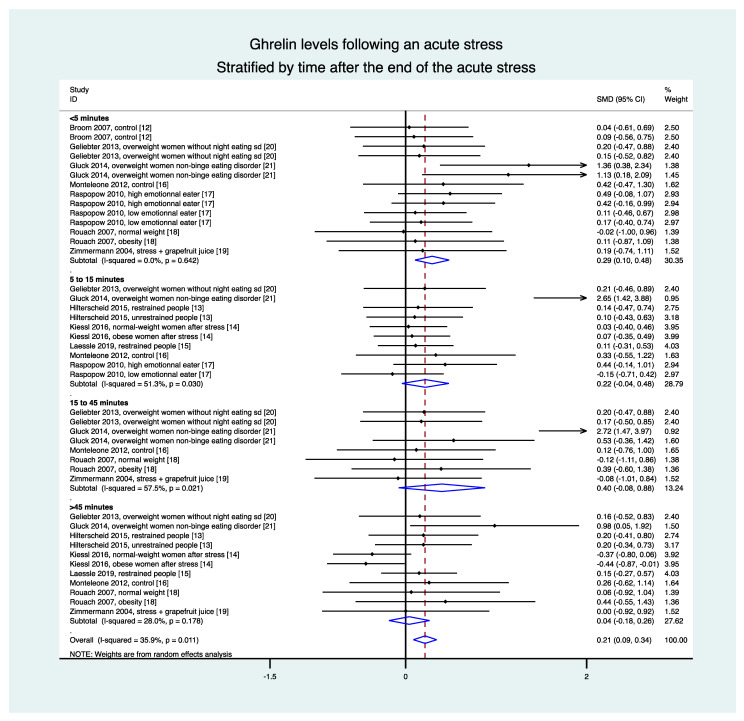
Meta-analysis of ghrelin levels following acute stress compared to baseline levels, stratified by time after the initiation of the acute stress intervention.

**Figure 4 nutrients-13-00784-f004:**
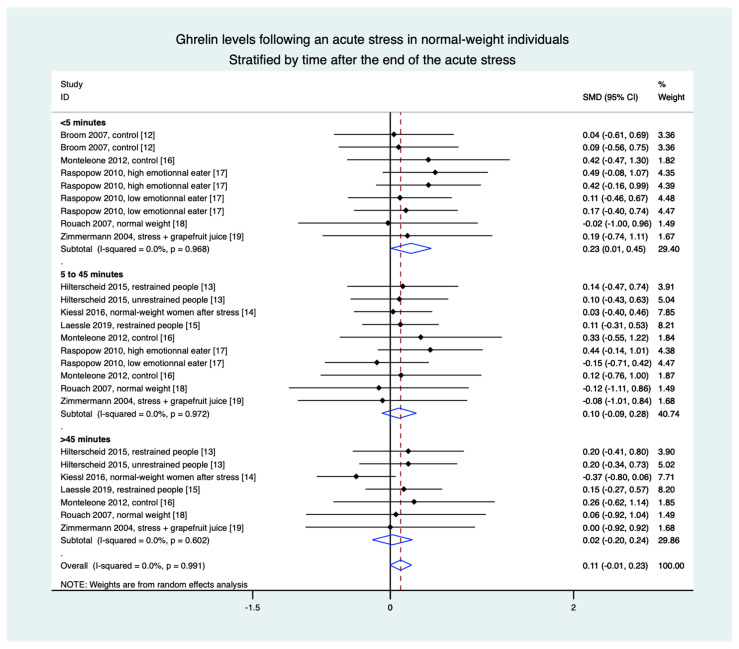
Meta-analysis of ghrelin levels after acute stress compared to baseline levels in the normal-weight population, stratified by time after the initiation of the acute stress intervention.

**Figure 5 nutrients-13-00784-f005:**
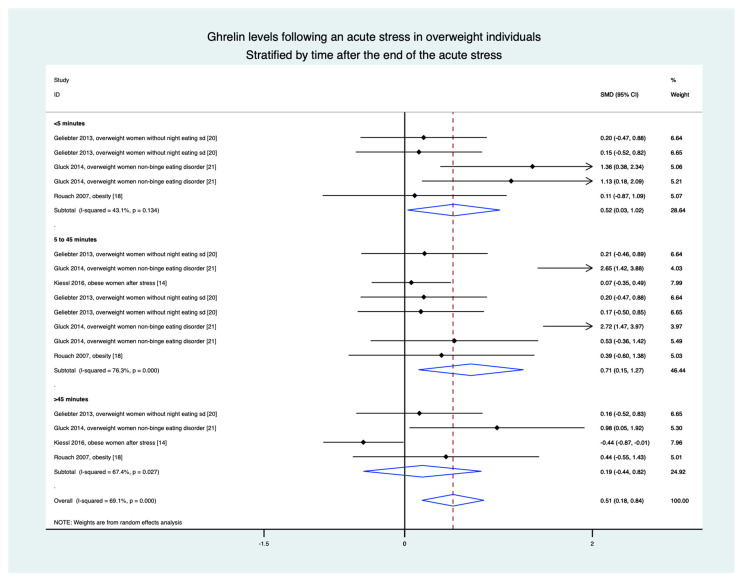
Meta-analysis of ghrelin levels after acute stress compared to baseline levels in the overweight population, stratified by time after the initiation of the acute stress intervention.

**Figure 6 nutrients-13-00784-f006:**
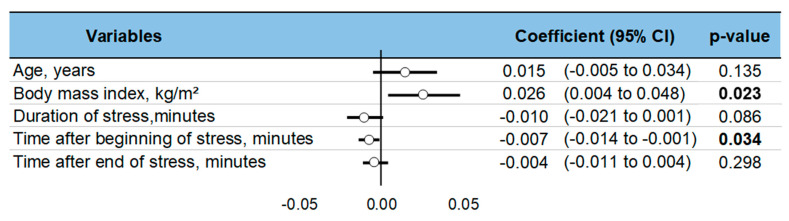
Meta-regressions—i.e., factors influencing changes in ghrelin levels following acute stress. Each variable’s effect on the outcome is represented in the forest-plot by a dot on a horizontal line. The circles represent the coefficient for each variable, and the length of each line around the dots represents their 95% confidence interval (95CI). The black solid vertical line represents the null estimate (with a value of 0). Horizontal lines that cross the null vertical line represent non-significant variables on the outcome. Bold *p*-value are significant.

**Figure 7 nutrients-13-00784-f007:**
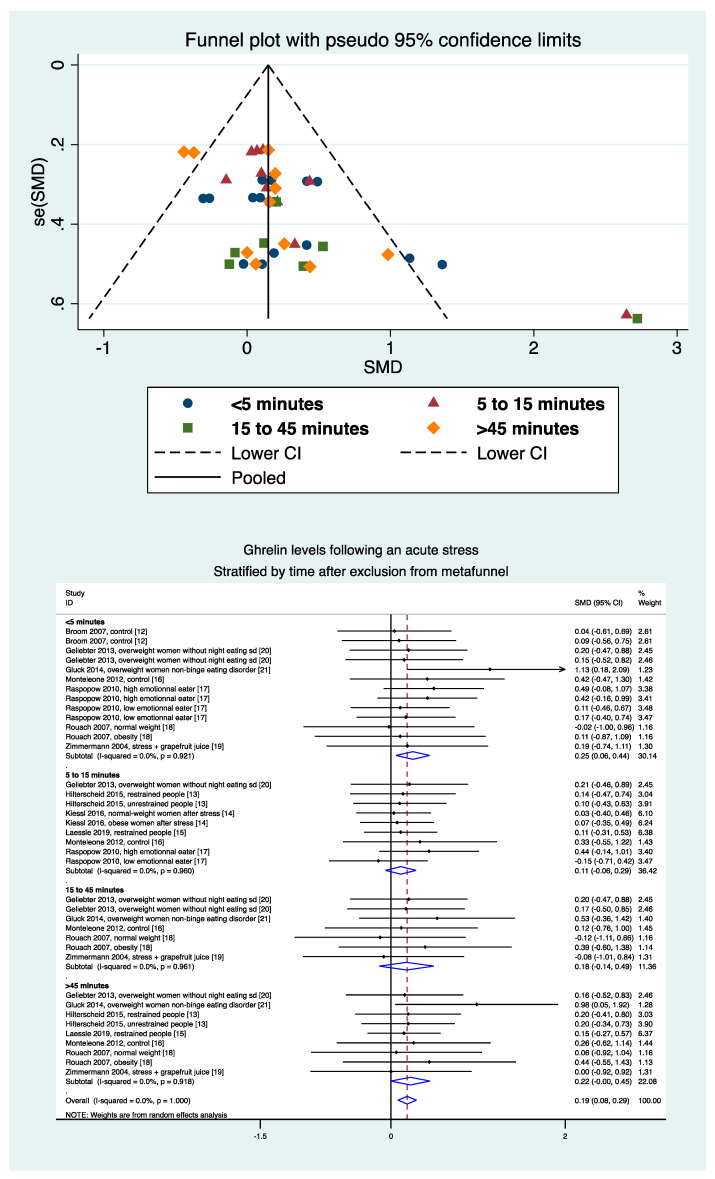
Metafunnel and meta-analysis of ghrelin levels after acute stress compared to baseline levels after exclusion from the metafunnel, stratified by time after the initiation of the acute stress intervention.

**Table 1 nutrients-13-00784-t001:** Descriptive characteristics of the included studies (all the studies had baseline measures of ghrelin levels, i.e., before stress).

Study	Country	Study Design	Population			Stress		Ghrelin Assessment		
			Characteristics	Men	Women	Type	Duration	Time after Stress	Fluid	Technique
				(*n*)	(*n*)		(min)	(min)		
Broom 2007 [12]	UK	Non-RCT	Training and controls	18	0	Physical (1 h at 75% VO_2_max)	60	2 measures: 0, 60	Blood	RIA
Geliebter 2014 [20]	USA	Non-RCT	Overweight women	0	17	Cold pressure test	2	6 measures: 0, 5, 15, 30, 45, 60	Blood	RIA
Gluck 2018 [21]	USA	Non-RCT	Overweight women	0	10	Cold pressure test	2	7 measures: 0, 2, 5, 15, 30, 45, 60	Blood	RIA
Hilterscheid 2015 [13]	Germany	Non-RCT	Women	0	27	Mental (TSST)	10	3 measures: 0, 30, 60	Blood	ELISA
Kiessl 2016 [14]	Germany	Non-RCT	Obese and normal-weight	0	85	Mental (TSST)	15	3 measures: 0, 30, 60	Blood	ELISA
Laessle 2018 [15]	Germany	Non-RCT	Women	0	48	Mental (TSST)	15	3 measures: 0, 30, 60	Blood	Not reported
Monteleone 2012 [16]	Italy	Non-RCT	Healthy women	0	10	Mental (TSST)	15	5 measures: 0, 10, 25, 40, 60	Saliva	ELISA
Raspopow 2012 [17]	Canada	Non-RCT	Women	0	48	Mental (TSST)	15	4 measures: 0, 10, 20, 30	Blood	RIA
Rouach 2007 [18]	Israël	Non-RCT	Obese and normal-weight	6	10	Mental (TSST)	15	5 measures: 0, 15, 35, 50, 75	Blood	RIA
Zimmermann 2006 [19]	Germany	Non-RCT	Healthy men	9	0	Mental (TSST)	15	5 measures: 0, 20, 35, 50, 65	Blood	RIA

CPT: Cold Pressor Test; ELISA: Enzyme-linked-immunosorbent-assay; RCT: Randomized Controlled Trial; RIA: Radio-immuno-assay; TSST: Trier Social Stress Test; UK: United Kingdom; USA: United States of America.

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
