# Peer review of "Ghrelin as a Biomarker of Stress: A Systematic Review and Meta-Analysis"

_nutrients, 2021, doi:10.3390/nu13030784_

Round 1

Reviewer 1 Report

Overall, this is a well-written paper with an interesting aim.

INTRODUCTION

The introduction must provide enough background information for readers to understand the problem, it would be recommended to increase this section to clarify the study aim.

METHODS

The experimental apparatus or equipment are quite standard. However, they are appropriate for the aim of the study.

More information about the specific data collection would be interesting. However, it is more than sufficient to reproduce the research.

RESULTS

Specify in tables the abbreviations.

DISCUSSION

All possible interpretations of the data considered are consistent.

Limitations are well established and comprehensive.

The conclusions have coherence with the initial hypothesis, in addition, they are well established and according to the present discussion.

LITERATURE CITED

The literature cited is relevant to the study

SIGNIFICANCE AND NOVELTY

As it stands, the results are novel and important enough for this journal.

Author Response

Reviewer 1:

Overall, this is a well-written paper with an interesting aim.

[REPLY] Thank you for your positive comment.

INTRODUCTION

The introduction must provide enough background information for readers to understand the problem, it would be recommended to increase this section to clarify the study aim.

[REPLY] Thank you for your positive comment. We added several sentences to explain deeper each point of the study aims (the need for biomarkers of stress; the conflicting results surrounding ghrelin response to acute stress; and the strong benefits that the scientific community may get from a better understanding of the relationship between stress and obesity, mediated by ghrelin response).

METHODS

The experimental apparatus or equipment are quite standard. However, they are appropriate for the aim of the study.

[REPLY] Thank you for your positive comment.

More information about the specific data collection would be interesting. However, it is more than sufficient to reproduce the research.

[REPLY] Thank you for your positive comment. We added some information and one sentence to justify the use of broad keywords. Whether you would like more information, please do not hesitate to ask.

RESULTS

Specify in tables the abbreviations.

[REPLY] Thank you for your positive comment. We added the following legend in Table 1: CPT: Cold Pressor Test; ELISA: Enzyme-linked-immunosorbent-assay; RCT: Randomized Controlled Trial; RIA: Radio-immuno-assay; TSST: Trier Social Stress Test; UK: United Kingdom; USA: United States of America.

DISCUSSION

All possible interpretations of the data considered are consistent.

[REPLY] Thank you for your positive comment.

Limitations are well established and comprehensive.

[REPLY] Thank you for your positive comment.

The conclusions have coherence with the initial hypothesis, in addition, they are well established and according to the present discussion.

[REPLY] Thank you for your positive comment.

LITERATURE CITED

The literature cited is relevant to the study

[REPLY] Thank you for your positive comment.

SIGNIFICANCE AND NOVELTY

As it stands, the results are novel and important enough for this journal.

[REPLY] Thank you for your positive comment.

Reviewer 2 Report

The authors have presented a well conducted systematic review with meta-analysis aiming to collate the available literature on the appetite regulation hormone ghrelin, and its involvement in the acute stress response.

The authors have conducted a rigorous systematic review, using appropriate methodology. The authors have stated that the search phrases used included “ghrelin or ghrelin level” and “stress or acute stress”. Did the authors consider using MeSH terms to conduct their search? This would have allowed authors to eliminate studies in animals, and in subpopulation groups within the search strategy itself which may have saved them time in excluding papers.

In section 3.3.4, the authors state that no studies reported leisure physical activity. One of the studies included in the meta-analysis recruited physically active or competitive athlete participants. Is this statement true, that physical activity was not reported even in this study?

Overall, the manuscript is well written and adds a novel contribution to the literature field. However, throughout the manuscript there are many malapropisms of the English language. For example, the word precluding has been misused throughout the manuscript. The sentence on line 207-209 “Insufficient data precluding further analyses by type of stress (mental, physical, or both), or by type of sampling (blood, saliva etc)”. I would suggest changing the suffix so that the sentence reads in the past tense; “Insufficient data precluded further analyses…”. A further example of this is on line 266 and 270, page 11.

Results are presented clearly and the text accompanies results reported in the forrest plots.

Thorough proofreading to ensure readability of the text is required. For example, sentence on lines 268-270 does not read well.

Section 4.4 discussing the prolonged ghrelin response in overweight and obese individuals warrants greater explanation. This is the arguably the most novel finding reported in this paper. I would have expected that the bulk of the discussion would be allocated to discuss the result that those who are overweight/obese have a higher and more prolonged ghrelin response to the acute stress. What are the potential biological mechanisms explaining this? Can you speculate as to why this might occur? Your explanation of the link between obesity and stress could be further developed to emphasise this finding and highlight what the study is contributing to the literature. It is important to acknowledge the direction of the relationship between the heightened and prolonged ghrelin response to stress in overweight and obese individuals. Do these individuals have a higher ghrelin response to stress, which contributed to the development of obesity? Or, have they developed a heightened ghrelin response as a result of weight gain. This is an important consideration which could be evidenced by further explanation of the biological mechanisms. It is also important because this has implications on how the findings of this study could be used in the real world. For example, if the ghrelin response to stress is heightened in some individuals, which results in them becoming overweight or obese, this could be an important screening tool to identify people who are at heightened risk of obesity.

Additionally, there are two sentences in a row (lines 280-284) which start with the same 7 words. Suggest a rewrite. The last sentence indicating stress management programs have been proposed for weight loss interventions is good, but this is also a section which could be further developed to clearly communicate why the findings of this paper are important. The authors could provide suggestions on how the novel finding of the link between ghrelin and stress, particularly in those who are overweight/obese, could benefit the population. For example, is there a way that appetite regulation or ghrelin testing could be used as part of weight management strategies to both promote weight loss or prevent weight gain? A section providing suggestions for the real world application of the findings forms a strong conclusion to the paper.

Author Response

Reviewer 2:

The authors have presented a well conducted systematic review with meta-analysis aiming to collate the available literature on the appetite regulation hormone ghrelin, and its involvement in the acute stress response.

[REPLY] Thank you for your positive comment.

The authors have conducted a rigorous systematic review, using appropriate methodology. The authors have stated that the search phrases used included “ghrelin or ghrelin level” and “stress or acute stress”. Did the authors consider using MeSH terms to conduct their search? This would have allowed authors to eliminate studies in animals, and in subpopulation groups within the search strategy itself which may have saved them time in excluding papers.

[REPLY] Thank you for your positive comment. We have chosen to use broad keywords to retrieve all possible articles.

In section 3.3.4, the authors state that no studies reported leisure physical activity. One of the studies included in the meta-analysis recruited physically active or competitive athlete participants. Is this statement true, that physical activity was not reported even in this study?

[REPLY] Thank you for your positive comment. Broom et al., in their study, recruited “sports science students studying at Loughborough University. All participants were physically active, and most were involved in competitive sports such as soccer, rugby, tennis, and hockey ». Furthermore, authors don’t report the quantity or quality of leisure physical activity.

Overall, the manuscript is well written and adds a novel contribution to the literature field. However, throughout the manuscript there are many malapropisms of the English language. For example, the word precluding has been misused throughout the manuscript. The sentence on line 207-209 “Insufficient data precluding further analyses by type of stress (mental, physical, or both), or by type of sampling (blood, saliva etc)”. I would suggest changing the suffix so that the sentence reads in the past tense; “Insufficient data precluded further analyses…”. A further example of this is on line 266 and 270, page 11.

[REPLY] Thank you for your positive comment. We sent our manuscript to a native English speaker who corrected many sentences, such as your examples. It became “Insufficient data precluded further analyses by type of stress (mental, physical, or both) or by type of sampling (blood, saliva, etc.)”

Results are presented clearly and the text accompanies results reported in the forrest plots.

[REPLY] Thank you for your positive comment.

Thorough proofreading to ensure readability of the text is required. For example, sentence on lines 268-270 does not read well.

[REPLY] Thank you for your positive comment. The manuscript has been proof read by a ntaive Eglish speaker.

Section 4.4 discussing the prolonged ghrelin response in overweight and obese individuals warrants greater explanation. This is the arguably the most novel finding reported in this paper. I would have expected that the bulk of the discussion would be allocated to discuss the result that those who are overweight/obese have a higher and more prolonged ghrelin response to the acute stress. What are the potential biological mechanisms explaining this? Can you speculate as to why this might occur?

[REPLY] Thank you for your positive comment. The following sentences have been added: “Although associations between ghrelin levels, obesity and stress exist, causal pathways have not been established.(32823562; 24860541) The biological pathways of ghrelin response to acute stress are yet to be fully understood (29927688), and therefore the biological explanations of a greater ghrelin response following an acute stress in obese individuals are also lacking.” as well as the beginning of the section that have been rewritten conforming to reviewer 3 suggestions. We welcome any suggestion that you may have. 

Your explanation of the link between obesity and stress could be further developed to emphasise this finding and highlight what the study is contributing to the literature. It is important to acknowledge the direction of the relationship between the heightened and prolonged ghrelin response to stress in overweight and obese individuals. Do these individuals have a higher ghrelin response to stress, which contributed to the development of obesity? Or, have they developed a heightened ghrelin response as a result of weight gain. This is an important consideration which could be evidenced by further explanation of the biological mechanisms. It is also important because this has implications on how the findings of this study could be used in the real world. For example, if the ghrelin response to stress is heightened in some individuals, which results in them becoming overweight or obese, this could be an important screening tool to identify people who are at heightened risk of obesity.

[REPLY] Thank you for your positive comment. The following sentences were added: “The direction of the relationship between the heightened and prolonged ghrelin response to stress in overweight and obese individuals would be important to acknowledge. It can be either a higher ghrelin response to stress contributing to the development of obesity, or obese individuals may have developed a heightened ghrelin response as a result of weight gain. Results from our studies may therefore have implications to use ghrelin as a screening tool to identify people at risk of obesity. Although associations between ghrelin levels, obesity and stress exist, causal pathways have not been established.(32823562; 24860541) The biological pathways of ghrelin response to acute stress are yet to be fully understood (29927688), and therefore the biological explanations of a greater ghrelin response following an acute stress in obese individuals are also lacking.”

Additionally, there are two sentences in a row (lines 280-284) which start with the same 7 words. Suggest a rewrite.

[REPLY] Thank you for your positive comment. We changed the second sentence to “It induces that international recommendations suggest implementing stress management programs in obesity for a sustainable weight loss.”

The last sentence indicating stress management programs have been proposed for weight loss interventions is good, but this is also a section which could be further developed to clearly communicate why the findings of this paper are important. The authors could provide suggestions on how the novel finding of the link between ghrelin and stress, particularly in those who are overweight/obese, could benefit the population. For example, is there a way that appetite regulation or ghrelin testing could be used as part of weight management strategies to both promote weight loss or prevent weight gain? A section providing suggestions for the real world application of the findings forms a strong conclusion to the paper.

[REPLY] Thank you for your positive comment. The following sentence has been added: “Appetite regulation testing warrant further studies to be used as part of weight management strategies, especially in overweight/obese individuals.”

Reviewer 3 Report

In this meta-analysis, the authors aimed to examine ghrelin levels following an acute stress and whether the response of ghrelin levels is higher in obese individuals compared to the normal-weight ones. In total, the authors included 10 studies for 348 participants. Type of acute stress comprised trier social stress test (7 studies), cold pressure test (2 studies), and maximum oxygen uptake treadmill running test (1 study). The overall meta-analysis showed an increased ghrelin levels after administration of acute stress (ES = 0.21, 95CI 0.09 to 0.34) compared to the baseline levels. Obese individuals had a more significant (ES = 0.51, 95CI 0.18 to 0.84) and prolonged increase in the ghrelin levels (up to 45 minutes) compared to non-obese ones (within 5 minutes). Insufficient data precluded further analyses by type of stress. The findings are interesting, and provide evidence for future research on ghrelin as a biomarker link between obesity and psychological stress. Below are my comments:

  1. Page 2, line 78: I did not see any supplementary materials. Please provide these data.
  2. Figure 3, 4, 5: Can the authors explain the repeated items shown in these figures? ex. Figure 3, < 5 minutes, two items of “Broom 2007, control”, etc.
  3. In Figure 4 which demonstrates the ghrelin levels following an acute stress in normal-weight individuals, the effects are small. Please discuss this in the manuscript.
  4. In Figure 5 which demonstrates the ghrelin levels following an acute stress in obese individuals, the I2 values are large. Please discuss this in the manuscript.
  5. Page 10, line 206 and 207: I did not see Figure 6 and 7. Please provide these figures.
  6. Page 10: Please indicate the position of Table 2 in the text.
  7. Page 12, line 293-296: I did not see any sensitivity analysis for type of stressors, sampling, and methods used to measure ghrelin levels.

Author Response

Reviewer 3:

In this meta-analysis, the authors aimed to examine ghrelin levels following an acute stress and whether the response of ghrelin levels is higher in obese individuals compared to the normal-weight ones. In total, the authors included 10 studies for 348 participants. Type of acute stress comprised trier social stress test (7 studies), cold pressure test (2 studies), and maximum oxygen uptake treadmill running test (1 study). The overall meta-analysis showed an increased ghrelin levels after administration of acute stress (ES = 0.21, 95CI 0.09 to 0.34) compared to the baseline levels. Obese individuals had a more significant (ES = 0.51, 95CI 0.18 to 0.84) and prolonged increase in the ghrelin levels (up to 45 minutes) compared to non-obese ones (within 5 minutes). Insufficient data precluded further analyses by type of stress. The findings are interesting, and provide evidence for future research on ghrelin as a biomarker link between obesity and psychological stress.

[REPLY] Thank you for your positive comment.

Below are my comments:

  1. Page 2, line 78: I did not see any supplementary materials. Please provide these data.

[REPLY] Thank you for your positive comment. We don’t have any supplementary materials. Sorry for that mistake. We deleted the sentence.

  1. Figure 3, 4, 5: Can the authors explain the repeated items shown in these figures? ex. Figure 3, < 5 minutes, two items of “Broom 2007, control”, etc.

[REPLY] Thank you for your positive comment. We added the following sentences in the Statistical considerations section: “All articles included reported several measurement time of ghrelin levels following acute stress. All meta-analysis were stratified by time after stress. The number of stratifications was determined by the number of studies within each stratification. For ex-ample, the main meta-analysis was stratified on 4 times of measures: <5 minutes, 5-15 minutes, 15-45 minutes, and >45 minutes.”

  1. In Figure 4 which demonstrates the ghrelin levels following an acute stress in normal-weight individuals, the effects are small. Please discuss this in the manuscript.

[REPLY] Thank you for your positive comment. We added the following sentences in section 4.4 “Even if we found a positive effect size in the overall population, there is a difference between normal-weight and overweight people. Indeed, the normal-weight population's effect size was small and only significant during the first five minutes. This could impact our conclusions on the overall meta-analysis and possibly reduce the generalization of our results. In contrast, we showed a higher and longer increase of ghrelin levels in the overweight patients than in the normal-weight patients, in accordance with the literature (43).”.

  1. In Figure 5 which demonstrates the ghrelin levels following an acute stress in obese individuals, the I2 values are large. Please discuss this in the manuscript.

[REPLY] Thank you for your positive comment. We added the following sentence in the section limitations. “The high I2 (69.1%) for meta-analysis in obese individuals might limit the generalization. However, we demonstrated a positive effect of BMI in our meta-regression.”

  1. Page 10, line 206 and 207: I did not see Figure 6 and 7. Please provide these figures.

[REPLY] Thank you for your positive comment. “Figure 6” was replaced by “Table 2”. The sentence is now “Sociodemographic (age, sex) did not influence the level of ghrelin, as well as the duration of stress or time of sampling (Table 2). Sensitivity analysis demonstrated similar results after exclusion of studies not evenly distributed around the funnel plot (Figure 6) ). Insufficient data precluded further analyses by type of stress (mental, physical, or both) or by type of sampling (blood, saliva, etc.)”. The figure 7 is now Figure 6 named “Metafunnel and meta-analysis of ghrelin levels after acute stress after exclusion of studies from the Metafunnel.

  1. Page 10: Please indicate the position of Table 2 in the text.

[REPLY] Thank you for your positive comment. Cf Response of your comment number 5. It was a mistake, and we corrected it.

  1. Page 12, line 293-296: I did not see any sensitivity analysis for type of stressors, sampling, and methods used to measure ghrelin levels.

[REPLY] Thank you for your positive comment. Indeed, as explained in the manuscript: “Insufficient data precluded further analyses by type of stress (mental, physical, or both) or by type of sampling (blood, saliva, etc.)”.

Round 2

Reviewer 2 Report

Thank you for addressing my comments. I have no further feedback or recommendations. 

Author Response

Reviewer 2:

Thank you for addressing my comments. I have no further feedback or recommendations.

[REPLY] Thank you for your positive comment.

Reviewer 3 Report

1. Page 10, line 214: “ES = 0.028, 95CI 0.006 to 0.49” was not identical to those shown in Table 2.

2. Page 12, line 325-326: Since there was no sensitivity analysis for type of stressors, sampling, and methods used to measure ghrelin levels, the sentence “no significant differences between the aforementioned methods in sensitivity analyses” is misleading.

Author Response

  1. Page 10, line 214: “ES = 0.028, 95CI 0.006 to 0.49” was not identical to those shown in Table 2.

[REPLY] Thank you for your positive comment. After the check of my Stata© program, I changed numbers in the main text. Those in Table 2 were good.

  1. Page 12, line 325-326: Since there was no sensitivity analysis for type of stressors, sampling, and methods used to measure ghrelin levels, the sentence “no significant differences between the aforementioned methods in sensitivity analyses” is misleading

[REPLY] Thank you for your positive comment. We changed the all sentence by “Moreover, homogeneity of data precluded further sensitivity analysis from different methods for measuring ghrelin levels (ELISA and RIA), stressors (TSST, sport, and CPT), and sampling (blood and saliva)”.
